# Performance Evaluation of Multiplex Molecular Syndromic Panel vs. Singleplex PCR for Diagnosis of Acute Central Nervous System Infections

**DOI:** 10.3390/microorganisms13040892

**Published:** 2025-04-13

**Authors:** Liliana Gabrielli, Miriam Tomaiuolo, Isabella Banchini, Alice Balboni, Andrea Liberatore, Federica Lanna, Alessia Cantiani, Alessia Bertoldi, Matteo Pavoni, Lamberto Manzoli, Tiziana Lazzarotto

**Affiliations:** 1Microbiology Unit, IRCCS Azienda Ospedaliero-Universitaria di Bologna, 40138 Bologna, Italy; miriam.tomaiuolo@studio.unibo.it (M.T.); andrea.liberatore@studio.unibo.it (A.L.); alessia.bertoldi3@unibo.it (A.B.); tiziana.lazzarotto@unibo.it (T.L.); 2Microbiology, Department of Medical and Surgical Sciences, University of Bologna, 40138 Bologna, Italy; isabella.banchini@studio.unibo.it (I.B.); alice.balboni2@studio.unibo.it (A.B.); federica.lanna@aosp.bo.it (F.L.); alessia.cantiani@studio.unibo.it (A.C.); matteo.pavoni2@unibo.it (M.P.); 3Department of Medical and Surgical Sciences, University of Bologna, 40138 Bologna, Italy; lamberto.manzoli2@unibo.it; 4Department of Environmental and Prevention Sciences, University of Ferrara, 44121 Ferrara, Italy

**Keywords:** meningitis, encephalitis, cerebrospinal fluid, molecular syndromic panel, viral infection

## Abstract

Acute central nervous system (CNS) infections, such as meningitis and encephalitis, represent medical emergencies that require rapid identification of the causative pathogen to guide appropriate therapeutic interventions. The QIAstat-Dx^®^ Meningitis/Encephalitis (QIA/ME) is a molecular syndromic panel that enables the simultaneous detection of multiple pathogens and provides the visualization of cycle threshold (Ct) values, offering rapid results for prompt clinical management. This study retrospectively tested, with the QIA/ME panel, 170 cerebrospinal fluid (CSF) samples from patients with CNS infections, confirmed through routine diagnostic workflows. The results were compared with those obtained from bacterial culture and singleplex PCR for viral detection. The QIA/ME demonstrated 100% concordance with reference methods for bacterial and yeast infections. For viral infections, the overall detection rate was 85.9%. Specifically, when singleplex PCR results exceeded 250 copies/mL for DNA viruses and 500 copies/mL for the RNA virus, the concordance rate with the QIA/ME was 96.8%. In contrast, when PCR values were below these thresholds, the concordance rate dropped to 43.8%. A strong overall correlation was observed between the viral load measured by singleplex PCR and Ct values from the QIA/ME (ρ = −0.83, *p* < 0.001). Only for enterovirus a weak correlation was found (ρ = −0.40, *p* = 0.056). The QIA/ME panel is an effective diagnostic tool for viral CNS infections, allowing for the visualization of Ct values that reflect pathogen load in samples and which could be useful in guiding clinical decision-making and patient management.

## 1. Introduction

Acute central nervous system (CNS) infections, such as meningitis and encephalitis, are medical emergencies that require prompt hospital care and rapid diagnosis of the etiological agent. These life-threatening infections can lead to severe, long-lasting debilitating sequelae [1,2,3].

Detection of bacterial pathogens in cerebrospinal fluid (CSF) requires a combination of techniques, including Gram staining (GS), antigen tests, and bacterial culture. For viral pathogens, identification using singleplex real-time PCR methods has demonstrated the highest sensitivity and specificity [2,4,5].

Rapid syndromic molecular assays for meningitis and encephalitis have been developed over the past few years [3,6].

In 2016, the first reports on the diagnostic application of the BioFire FilmArray^®^ Meningitis/Encephalitis (ME) panel (FA/ME; BioFire Diagnostics, bioMérieux LLC, Salt Lake City, UT, USA) were published [1,7]. The main advantage of multiparametric assays is that they encompass bacteria, viruses, and yeast, covering the most common causative agents of infectious meningitis/encephalitis in approximately one hour. The reduced time to obtain a diagnostic result has improved antimicrobial and antiviral treatment, enabling de-escalation of empiric therapy [3,6,8,9]. The available literature suggests that the FA/ME has high diagnostic accuracy, with a sensitivity of 90% and specificity of 97% [8,10]. False-negative results have been reported, particularly for herpes simplex virus (HSV) [10,11] and enterovirus (EV) [8,10]. In cases where the syndromic panel is negative but there remains a high clinical suspicion of viral infections, singleplex PCR and evaluation of other sample types (e.g., blood and/or vesicle swabs) should be considered [2,10,12].

Recently, a syndromic panel that provides amplification curves and cycle threshold (Ct) values for qualitative and semi-quantitative evaluation has become available: the QIAstat-Dx^®^ Meningitis/Encephalitis panel (QIA/ME; QIAGEN, Hilden, Germany) [5,13,14,15,16]. This panel allows for more detailed analysis compared to previous systems.

In the current study, the QIA/ME was used to evaluate 170 CSF samples that had previously tested positive for one or more of the QIA/ME targets using reference methods, in a Northern Italian University Hospital laboratory.

The main objective of this study was to assess the clinical utility of the QIA/ME syndromic panel in viral CNS infections, focusing on the sensitivity of the test and the linear correlation between the viral load obtained by the reference method and the Ct values displayed by the QIA/ME panel. The investigation was conducted using retrospective residual CSF specimens from patients who presented with signs and symptoms of meningitis and/or encephalitis.

## 2. Materials and Methods

### 2.1. Clinical Samples

CSF specimens from patients with suspected CNS infections, confirmed by routine diagnostic workflows, were analyzed using the QIA/ME panel from August to October 2024 at the Laboratory of Virology, Operative Unit of Microbiology, IRCCS Sant’Orsola Policlinic of Bologna. The CSF specimens included in this study met the following criteria: they were collected from patients with clinical signs of meningitis and/or encephalitis and were residual samples from the routine diagnostic workflow that yielded positive results for one or more of the QIA/ME targets. The specimens had a total volume of up to 800 μL and were stored at −80 °C for no more than 5 years, without undergoing any freeze–thaw cycles. The samples were anonymized prior to freezing, and no clinical data were available.

No positive samples were found for *E. coli* K1 and *M. pneumoniae*. A total of 170 CSF samples, all positive by the reference method, were analyzed.

### 2.2. Routine Diagnostic Workflow

The routine diagnostic workflow includes biochemical evaluation, GS, culture testing, and real-time PCR for viral detection [2]. The total volume of CSF should be approximately 8 mL in adults (with a minimum volume of 1 mL), which is divided into three separate samples. The first sample, approximately 2 mL, is used for biochemical analysis; the second for culture testing, microscopic examination, and direct detection of microbial antigens; and the third for viral investigations via molecular routine tests.

CSF specimens were prepared for GS and culture tests by centrifugation at 3000 rpm for 10 min at room temperature. For GS, 1–2 drops of the well-mixed CSF sediment were placed on a slide to create a smear. After drying, the smear was fixed, stained, and observed under light microscopy with a 100× objective.

Culture testing was performed by seeding 1–2 drops of CSF sediment directly onto the following agar plates: horse-blood agar, Thayer–Martin agar, chocolate agar, and Sabouraud agar. Additionally, a brain–heart infusion broth tube was inoculated with one drop of the sediment. The agar plates and broth were incubated for up to 5 days at 35–37 °C (~5% CO_2_, or in a candle jar for Thayer–Martin and chocolate agar). Positive cultures were identified using MALDI-TOF mass spectrometry (Bruker Daltonics, Milan, Italy). In addition to GS and culture, a rapid and reliable identification of *S. pneumonia* was performed using a fluorescence immunoassay (STANDARD F *S. pneumoniae* Ag FIA, SD Biosensor, Gyeonggi, Republic of Korea) with an automated reader (STANDARD F2400 Analyzer, SD Biosensor, Gyeonggi, Republic of Korea), which detects the C polysaccharide cell wall antigen common to all pneumococcal serotypes in CSF samples before centrifugation for microscopic and culture examination. *C. neoformans* was detected by the India ink test, culture, and latex agglutination test (LAT). The LAT assay was performed on 25 μL of CSF using the Latex-Cryptococcus antigen detection system kit (IMMY, Norman, OK, USA), according to the manufacturer’s instructions.

For viral investigations, molecular routine tests were performed on CSF to identify and quantify HSV-1, HSV-2, varicella zoster virus (VZV), human herpesvirus 6 (HHV-6), EV, and cytomegalovirus (CMV). The tests used in routine diagnostics involved the ELITeInGenius™ system and ELITe MGB Kits (ELITechGroup S.p.A., Torino, Italy). The ELITeInGenius™ system is a fully automated instrument: DNA extraction from 200 µL of CSF was performed using ready-to-use cartridges, and real-time PCR was conducted in dedicated cartridges for each sample. An internal control (IC) was included during the extraction phase to verify the absence of PCR inhibitors. Specific, single-target, quantitative real-time ELITE MGB Kits were used for the virus detection and quantification. The system automatically analyzes and interprets the results. Each Elite MGB kit has specific Limit of Quantification (LoQ) in CSF—which corresponded to the Lower Limit of Detection (LoD). The LoQ were 250 copies/mL for HSV-1, 119 copies/mL for HSV-2, 69 copies/mL for VZV, 250 copies/mL for HHV-6, and 490 copies/mL for EV. To simplify the reporting to clinicians, a Laboratory LoQ (LaLoQ) with the InGenius system was conservatively set at 500 copies/mL for EV and 250 copies/mL for HSV-1, HSV-2, VZV, and HHV-6. Parechovirus (HPeV) was detected using the Meningitis Viral 2 ELITe MGB Panel, a multiplex qualitative real-time PCR assay.

### 2.3. QIAstat-Dx^®^ Meningitis/Encephalitis Panel

QIA/ME is a CE-IVD-marked multiplex real-time PCR assay that uses a cartridge-based system. The test was conducted following the manufacturer’s instructions. Briefly, 200 μL of CSF was transferred into the main port of the cartridge with the supplied transfer pipettes. Once the test was initiated, the QIAstat-Dx Analyzer 1.0 automatically handled the extraction, amplification, and detection of nucleic acids. The QIA/ME cartridge contains an IC to monitor all steps of the process. If the IC signal is absent, negative results are considered invalid. Test results were available in approximately 80 min. Individual LoD values for each viral target are reported in Appendix A.

### 2.4. Statistical Analysis

The linear correlation between the viral load, obtained using the singleplex molecular test, and the Ct values of the syndromic panel was evaluated using statistical analysis of Pearson’s ρ coefficient (a value close to −1 or +1 suggests a strong correlation). A *p*-value less than 0.05 was considered as a statistical significance cut-off. Confidence intervals (CIs) were computed at the 95% level. Data falling below the LaLoQ were left-censored as 249 copies/mL for the results <250 copies/mL and as 499 copies/mL for the results < 500 copies/mL.

All tests and statistical analyses were performed using Stata version 11.

## 3. Results

### 3.1. Clinical Specimens

A total of 170 CSF samples were analyzed, all of which tested positive for viruses, bacteria, or yeast using routine diagnostic reference methods. Among these, 20 CSF samples were positive for yeasts or bacteria: 2 for *C. neoformans*, 1 for *H. influenzae*, 1 for *S. pyogenes*, 2 for *L. monocytogenes*, 2 for *S. agalactiae*, 3 for *N. meningitidis*, and 9 for *S. pneumoniae*. All 20 bacterial/yeast-positive results were available for comparison between the two methods. The remaining 150 CSF samples tested positive for viral pathogens. There were 11 coinfections, of which 8 were viral coinfections (3 positive for HHV-6 and VZV, 2 for VZV and HSV-1, 1 for HSV-2 and HHV-6, 1 for EV and HHV-6, and 1 for HSV-1 and HHV-6), and 3 were viral-bacterial/yeast coinfections (1 for HHV-6 and *S. agalactiae*, 1 for HHV-6 and *H. influenzae*, and 1 for HSV-1 and *C. neoformans*). Accordingly, 158 viral-positive results were available for comparison between the two methods: 36 positive for HSV-1, 15 for HSV-2, 40 for VZV, 28 for EV, 37 for HHV-6, and 2 for HPeV. The routine real-time PCR for HPeV is a qualitative method; therefore, the correlation between the viral load detected by singleplex PCR and Ct value by the QIA/ME was not available for the two HPeV-positive CSF samples. Consequently, 156 viral-positive results were available for evaluating the correlation between the viral load and Ct value.

### 3.2. Performance of QIA/ME vs. Reference Method

All 20 CSF samples positive for bacteria or yeast, as detected by the reference methods, were confirmed by the QIA/ME, yielding a 100% concordance rate. The median Ct value for these samples was 26.3, with a range from 13.9 to 35.7 (Appendix A). Among the 158 viral-positive results, the two HPeV-positive cases were confirmed. Regarding the 156 viral-positive results obtained using the quantitative reference method (singleplex PCR), the QIA/ME yielded positive results in 134 cases, corresponding to an overall detection rate of 85.9% (Appendix A).

Table 1 presents the comparison of positive results between the singleplex PCR and QIA/ME, showing a strong overall correlation (ρ = −0.83, *p* < 0.001). Among the 22 viral-negative results with the QIA/ME, 18 had a viral load below the LaLoQ, and, in four cases, the viral load was slightly above the LaLoQ. Specifically, two CSF samples yielded positive results for EV-RNA (1312 copies/mL and 4939 copies/mL), one for VZV (411 copies/mL), and one for HSV-1 (1143 copies/mL). In cases where singleplex PCR results exceeded the LaLoQ, the proportion of concordant positive results with the QIA/ME was 96.8% (95% CI: 91.9–99.1). In cases where PCR values were below the LaLoQ, the proportion of concordant positive results was 43.8% (95% CI: 26.4–62.3), as shown in Table 1.

### 3.3. Correlation Between Viral Load Detected by Singleplex PCR (Routine Method) and Ct Values Provided by QIA/ME

A total of 156 cases were evaluated for the correlation between the viral load and Ct values. The two HPeV-positive CSF samples were excluded from the analysis as the routine PCR for HPeV is a qualitative method. Figure 1 illustrates this correlation, considering each virus individually. A statistically significant correlation (*p* < 0.001) was observed for HSV-1, HSV-2, VZV, and HHV-6, while a weak correlation (*p* = 0.056) was found for EV.

## 4. Discussion

Acute CNS infections represent a major medical concern as early diagnosis and prompt therapeutic intervention are crucial for patient outcomes [17].

The primary objective in the early management of acute CNS infections is the rapid differentiation between various etiologies. Given that treatment is highly dependent on the underlying cause of the disease, identifying the responsible pathogen is essential to initiate the appropriate antimicrobial therapy [9]. In immunocompetent patients presenting with suspected meningitis or encephalitis, the routine diagnostic workflow includes an assessment of cellular and biochemical parameters, GS, bacterial culture, and singleplex PCR for EV, VZV, HSV-1, and HSV-2. In immunocompromised patients, additional viral-specific PCR assays are performed, such as HHV-6 for hematologic patients and CMV for transplant recipients. In cases with suspected cryptococcal meningitis, the detection of *C. neoformans* antigen is performed [18]. In pediatric patients, HHV-6 PCR and HPeV PCR can also be investigated [19].

Challenges in the microbiological diagnosis of CNS infections include reduced sensitivity of GS and culture yields when empirical antibiotics are administered prior to lumbar puncture, as well as a prolonged turnaround time (TAT) for microbiological results, which can range from several hours to up to two days [2,19]. Since 2016, several studies have investigated the diagnostic application of the syndromic PCR panels such as the FA/ME, which simultaneously detects 14 pathogens involved in CNS infections. A systematic review [8] evaluated the diagnostic accuracy of this panel by pooling data from eight studies comprising 3059 patients. In meta-analysis, the mean sensitivity was found to be 90% (95% CI 86–93%), while the mean specificity was 97% (95% CI 94–99%), with HSV-1, HSV-2, EV, and *C. neoformans/gattii* exhibiting the highest rates of false-negative results. The syndromic PCR offers multiple advantages over singleplex PCR, such as the ability to detect multiple microorganisms simultaneously with a reduced TAT.

In 2022, the QIA/ME was introduced. The main difference between the two syndromic tests is that the QIA/ME does not include CMV detection but does detect *S. pyogenes* and *M. pneumoniae*. Furthermore, it allows the visualization of amplification curves and Ct values, providing both qualitative and semi-quantitative evaluation [13]. A recent clinical trial [14], involving three European testing sites, demonstrated that the QIA/ME exhibits similar performance to the FA/ME. In particular, HSV-1 and *C. neoformans/gattii* were detected equally by both platforms, suggesting the same sensitivity issues regarding these targets.

The aim of our study was to assess the sensitivity of the QIA/ME using singleplex PCR (considered the gold standard) as a comparator for detecting the more frequent viral targets. We found that the sensitivity was lowest for HSV-1 (80.6%), whereas sensitivity for HSV-2 was 100%. In the literature, it is reported that HSV-1 is the main pathogen involved in false-negative results by syndromic tests [8,10,20]. Notably, false-negative PCR results for HSV-1 are reported in the early stages of encephalitis, even with singleplex PCR. Consequently, clinical guidelines for the diagnosis and management of encephalitis recommend that in cases with negative PCR for HSV-1 but a clinical syndrome compatible with encephalitis or evidence of temporal lobe lesions on neuroimaging, antiviral treatment should be continued, and repeat PCR testing should be performed after 3–7 days. In such instances, singleplex PCR should be utilized if the syndromic panel test is negative, given the lower sensitivity of the latter.

In our lab, to simplify reporting to clinicians, the LaLoQ for the singleplex PCR system was conservatively set at 250 copies/mL for all DNA viruses and 500 copies/mL for EV. Interestingly, we observed that CSF samples with a viral load below the LaLoQ by singleplex PCR yielded a positive result by syndromic testing in only 43.8% of cases.

Differently, the sensitivity of the syndromic test for detecting samples with viral loads above the LaLoQ of singleplex PCR was 96.8% for all viral targets. Specifically, the highest concordance was observed for HSV-2 and HHV-6 (100%), while the lowest was seen for EV (90.9%), as shown in Table 1. Among the four discordant samples with viral loads slightly above the LaLoQ, half were positive for EV. Given the large number of types in EV family, a less efficient detection of some of them is reported; in particular, the QIA/ME is not able to detect the lowest loads of E-30 and EV-D68-B3 types [15].

Additionally, the QIA/ME is able to identify coinfections. In our study, we evaluated 11 samples with coinfections, including eight viral coinfections, and three were viral-bacterial/yeast coinfections, and the QIA/ME successfully detected all the pathogens involved. The ability of only one molecular test to simultaneously identify multiple pathogens—bacterial, viral, and fungal agents—represents a significant advantage of this panel.

Considering both bacterial and yeast infections, all 20 CSF samples that tested positive using the reference methods were also positive by the QIA/ME, demonstrating a 100% concordance rate. The literature reports that an advantage of using molecular tests is their ability to provide accurate results in patients who have received antimicrobial therapy prior to lumbar puncture as negative results obtained with conventional methods, such as bacterial culture, may occur in such cases [3].

To our knowledge, this is the first study to investigate the correlation between Ct values and viral loads detected by singleplex PCR in a large cohort of samples using the QIA/ME. From the 156 CSF samples analyzed, we demonstrated a significant correlation between Ct values and viral load (ρ = −0.83; *p* < 0.001). Stratification by pathogen revealed the highest correlation for HSV-1 (ρ = −0.93), followed by HSV-2 and VZV (ρ = −0.92), HHV-6 (ρ = −0.85), and EV (ρ = -0.40). The poor correlation observed between the EV viral load and Ct values, with no statistical significance (*p* = 0.056), may be attributed to the differential detection efficiency for various EV types, as previously discussed. Regarding the stability of CSF samples stored at −80 °C for up to five years, although RNA is generally more prone to degradation than double-stranded DNA, we do not believe this storage condition significantly impacted the viral load. Stability studies on influenza virus by Granados et al. [21] have demonstrated no difference in viral load when clinical specimens were stored at −80 °C for up to three years and re-extracted. Similarly, José et al. [22] reported no decrease in hepatitis C virus load in unextracted serum samples stored at −70 °C for up to five years.

Currently, the correlation between viral load and clinical severity, the duration of symptoms, and sequelae is poorly studied. Concerning HSV-1 encephalitis, Bhullar et al. [23] suggested that higher HSV-1 loads in CSF correlate with more severe neurological impairment, including brain lesions detected by imaging, a longer duration of antiviral therapy, and poorer clinical outcomes. Therefore, the use of a panel able to allow Ct value visualization, reflecting the pathogen load in samples, could be useful in guiding clinical decision-making and patient management.

## 5. Conclusions

Multiparametric molecular panels are a valuable tool for the rapid and accurate diagnosis of meningitis and encephalitis. Compared to traditional diagnostic methods, such as culture and quantitative molecular tests, they offer several advantages, including the simultaneous detection of multiple pathogens and a reduced TAT. However, they have certain limitations, including the lack of information on antimicrobial susceptibility in bacterial infections, the exclusion of some potential CNS pathogens (e.g., Toscana virus and West Nile virus), and lower sensitivity (approximately 90%) compared to singleplex PCR. Furthermore, they do not provide quantitative data (copies/mL).

In our study, the QIA/ME demonstrated an overall sensitivity of 85.9% for detecting viral infections. When singleplex PCR results exceeded 250 copies/mL for DNA viruses and 500 copies/mL for the RNA virus, the concordance rate with QIA/ME was 96.8%. Additionally, we observed a strong correlation (ρ = −0.83, *p* < 0.001) between Ct values and viral load, providing valuable insights.

## Figures and Tables

**Figure 1 microorganisms-13-00892-f001:**
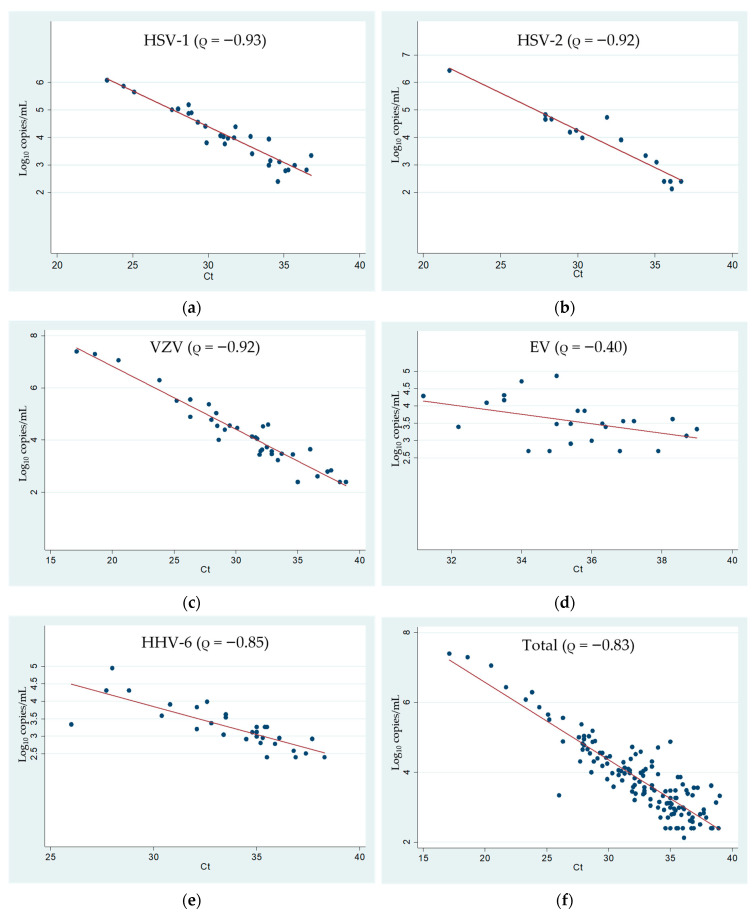
(**a**–**e**) Correlation between viral load (singleplex PCR) and Ct values (QIA/ME); (**f**) correlation between overall viral load (singleplex PCR) and Ct values (QIA/ME).

**Table 1 microorganisms-13-00892-t001:** Comparison between singleplex PCR and QIA/ME results by pathogen.

Pathogen	Positive Samples, Overall % (95% CI) ^A^	(n/N)	ρ ^D^	*p* ^E^	Positive Samples, >LaLoQ % (95% CI) ^B^	Positive Samples, <LaLoQ % (95% CI) ^C^
HSV-1	80.6 (64.0–91.8)	(29/36)	−0.93	<0.001	96.6 (82.2–99.9)	14.3 (0.3–57.9)
HSV-2	100.0 (78.2–100.0)	(15/15)	−0.92	<0.001	100.0 (73.5–100.0)	100.0 (29.2–100.0)
VZV	90.0 (76.3–97.2)	(36/40)	−0.92	<0.001	97.1 (84.7–99.9)	50.0 (11.8–88.2)
EV	85.7 (67.3–96.0)	(24/28)	−0.40	0.056	90.9 (70.8–98.9)	66.7 (22.3–95.7)
HHV-6	81.1 (64.8–92.0)	(30/37)	−0.85	<0.001	100.0 (87.2–100.0)	30.0 (6.7–65.2)
Total	85.9 (79.4–90.9)	(134/156)	−0.83	<0.001	96.8 (91.9–99.1)	43.8 (26.4–62.3)

n—number of concordant samples; N—total number of samples. ^A^ Proportion of positive (concordant) samples according to QIA/ME, overall sample. ^B^ Proportion of positive (concordant) samples according to QIA/ME, only the samples with PCR values higher than the LaLoQ. ^C^ Proportion of positive (concordant) samples according to QIA/ME, only the samples with PCR values lower than the LaLoQ. ^D^ Spearman ρ of the correlation between Ct QIA/ME values and singleplex PCR values (Log_10_ transformed). ^E^ *p*-value of the correlation between Ct QIA/ME values and singleplex PCR values (Log_10_ transformed).

## Data Availability

The original contributions presented in this study are included in the article/Appendix A. Further inquiries can be directed to the corresponding author.

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
