# Peer review of "Performance Evaluation of Multiplex Molecular Syndromic Panel vs. Singleplex PCR for Diagnosis of Acute Central Nervous System Infections"

_microorganisms, 2025, doi:10.3390/microorganisms13040892_

Round 1
Reviewer 1 Report
Comments and Suggestions for Authors
The study is quite interesting from a practical point of view, although its scientific significance is negligible. The slightly lower sensitivity of the new technique compared to the reference ones is rewarded by the simultaneous determination of many possible etiological factors of CNS infections.
Unfortunately, it lacks some clinically important pathogens, e.g. EBV or TBEV. There is no information on the sensitivity of reference PCR techniques for each virus, nor on the sensitivity limit of the multiplex method (copy/mL, or genomEq/mL), e.g. based on serial dilutions of the appropriate targets. This significantly limits the assessment of the value of the described diagnostic kit.
A collection of CSF samples in which the etiological agent of neuroinfection was not detected by routine techniques should also be tested using the described method and presented as part of the study. It is possible that one or more pathogens could be detected in these samples.
Fig. 1. There is no PCR viral load description on the vertical axis of the graph.
Author Response
Thank you very much for taking the time to review this manuscript. Please find the detailed responses below and the corresponding revisions highlighted in the re-submitted files.
Point-by-point response to Comments and Suggestions for Authors
Comments 1: The study is quite interesting from a practical point of view, although its scientific significance is negligible. The slightly lower sensitivity of the new technique compared to the reference ones is rewarded by the simultaneous determination of many possible etiological factors of CNS infections. Unfortunately, it lacks some clinically important pathogens, e.g. EBV or TBEV.
Response 1: The QIAstat-Dx ME Panel Cartridge enables the detection of 15 bacterial, viral, and fungal pathogens associated with meningitis and/or encephalitis. The pathogens that can be detected and identified using the QIAstat-Dx ME Panel include: Escherichia coli K1, Haemophilus influenzae, Listeria monocytogenes, Neisseria meningitidis (encapsulated), Streptococcus agalactiae, Streptococcus pneumoniae, Mycoplasma pneumoniae, Streptococcus pyogenes, Herpes simplex virus 1, Herpes simplex virus 2, Human herpesvirus 6, Enterovirus, Human parechovirus, Varicella-zoster virus, and Cryptococcus neoformans/gattii.
Other pathogens, such as EBV or TBEV, are not included; therefore, we were unable to evaluate these pathogens.
Comments 2: There is no information on the sensitivity of reference PCR techniques for each virus, nor on the sensitivity limit of the multiplex method (copy/mL, or genomEq/mL), e.g. based on serial dilutions of the appropriate targets. This significantly limits the assessment of the value of the described diagnostic kit.
Response 2: Thank you for this valuable suggestion. We have incorporated the requested changes into the text. Specifically:
- 2. Materials and Methods – Section 2.2 (page number 3):
We added the following (lines 127-128; 131-139): "Specific, single-target, quantitative real-time ELITE MGB Kits were used for virus detection and quantification. Each ELITE MGB kit has a specific Limit of Quantification (LoQ) in CSF, which corresponds to the Lower Limit of Detection (LoD). The LoQs were as follows: 250 copies/mL for HSV-1, 119 copies/mL for HSV-2, 69 copies/mL for VZV, 250 copies/mL for HHV-6, and 490 copies/mL for EV.
To simplify reporting to clinicians, a Laboratory LoQ (LaLoQ) using the InGenius system was conservatively set at 500 copies/mL for EV and 250 copies/mL for HSV-1, HSV-2, VZV, and HHV-6.
Parechovirus (HPeV) was detected using the Meningitis Viral 2 ELITE MGB Panel, a multiplex qualitative real-time PCR assay."
- 2. Materials and Methods – Section 2.3 (page number 4):
We added a specific section regarding the QIA/ME method (lines 141-150). This section provides a brief description of the method.
Individual LoD values for each viral target in the QIAstat-Dx ME Panel are reported in Table S1 of the supplementary section.
Comments 3: A collection of CSF samples in which the etiological agent of neuroinfection was not detected by routine techniques should also be tested using the described method and presented as part of the study. It is possible that one or more pathogens could be detected in these samples.
Response 3: The primary objective of this study was to evaluate the clinical utility of the QIA/ME syndromic panel in viral CNS infections, with a focus on the sensitivity of the test and the linear correlation between viral load determined by the reference method and the Ct values obtained from the QIA/ME panel. Negative samples were not investigated by the reference method, as it was considered the gold standard.
Comments 4: Fig. 1. There is no PCR viral load description on the vertical axis of the graph.
Response 4: Thank you for this clarification. It was a layout issue, and it has now been corrected.
Additional corrections
- The registered trademark symbol “®” has been added where necessary. Specifically, we used "BioFire FilmArray®" (lines 54), instead of "BioFire FilmArray" and "QIAstat-Dx®" (lines 20, 69) instead of "QIAstat-Dx."
- The abbreviated form for parechovirus, HPeV, has been used where necessary.
- The layout of Figure 1 (page number 6) has been modified to comply with the MDPI figure template.
- In the Abstract (page number 1): lines 29-30; line 31; lines 36-37).
- In the Introduction (page number 2): line 69.
- In Materials and Methods: section 2.1. (page number 2; line 91); section 2.2. (page number 3; lines 127-139).
- In the Discussion (page number 7): lines 271-272.
- In the Conclusions (page number 9): lines 327-330.
- “LaLoQ” has been used instead of “LLoQ” of where necessary.
- “Log10” has been used instead of “log(10)” (lines 204 and 206).
- Supplementary Materials section (page number 9, lines 334-343).

Reviewer 2 Report
Comments and Suggestions for Authors
Liliana Gabrielli,et al. compared the performance of QIA/ME and single PCR in the diagnosis of acute central nervous system infections (such as meningitis and encephalitis), which has certain guiding significance for the rapid clinical screening of acute central nervous system infections. However, the sample background information and test results are not clear, lack of sensitivity of QIA/ME method, detection threshold, etc.
1. Supplement QIA/ME method principle, sensitivity of each detection target, detection critical line;
2. Figure 1. A-F lacks abscissa and ordinate headings and units;
3. It is suggested to supplement the sample background information and test results in the form of supplementary data, and list the gold standard, reference method, QIA/ME and single PCR test results respectively. 170 samples of cerebrospinal fluid, all with a single pathogen? Are there samples of mixed infection?
Author Response
Thank you very much for taking the time to review this manuscript. Please find the detailed responses below and the corresponding revisions highlighted in the re-submitted files.
Point-by-point response to Comments and Suggestions for Authors
Comments 1: Supplement QIA/ME method principle, sensitivity of each detection target, detection critical line.
Response 1: Thank you for this valuable suggestion. We have incorporated the requested changes into the text. Specifically:
- 2. Materials and Methods – Section 2.3. (page number 4):
We added a dedicated section describing the QIA/ME method (lines 141-150). This section provides a brief overview of the method.
Individual LoD values for each viral target in the QIAstat-Dx Panel are reported in Table S1 in the supplementary section.
Comments 2: Figure 1. A-F lacks abscissa and ordinate headings and units.
Response 2: Thank you for this clarification. It was a layout issue, and it has now been corrected.
Comments 3: It is suggested to supplement the sample background information and test results in the form of supplementary data, and list the gold standard, reference method, QIA/ME and single PCR test results respectively.
170 samples of cerebrospinal fluid, all with a single pathogen? Are there samples of mixed infection?
Response 3: In the Materials and Methods paragraph– section 2.1. (page number 2, lines 85-90), we had already specified all the available information, as the samples were anonymized prior to freezing, and no clinical data were available. Specifically, we had already included in the first submission the following statement:
“The CSF specimens included in this study met the following criteria: they were collected from patients with clinical signs of meningitis and/or encephalitis and were residual samples from the routine diagnostic workflow that yielded positive results for one or more of the QIA/ME targets. The specimens had a total volume of up to 800 μL and were stored at −80°C for no more than 5 years, without undergoing any freeze-thaw cycles.”
As requested, we have added the overall results in the supplementary section, including 11 tables (from Table S.2 to Table S.12). The results were categorized as concordant and discordant for each virus. The results for bacteria and yeast were summarized in a single table, as all were concordant. CSF samples positive for bacteria/yeast are identified by the sample code “QIAMEBXX”, while CSF samples positive for viruses are identified by the sample code “QIAMEVXXX”.
Regarding mixed infections, we had already stated in the first submission, in the Results paragraph – section 3.1. (page number 4, lines 167-171):
“The remaining 150 CSF samples tested positive for viral pathogens. There were 11 coinfections, of which 8 were viral coinfections (3 positive for HHV-6 and VZV, 2 for VZV and HSV-1, 1 for HSV-2 and HHV-6, 1 for EV and HHV-6, and 1 for HSV-1 and HHV-6), and 3 were viral-bacterial/yeast coinfections (1 for HHV-6 and S. agalactiae, 1 for HHV-6 and H. influenzae, and 1 for HSV-1 and C. neoformans).”
In the supplementary material, all coinfections are clearly identified. Specifically, viral coinfections can be recognized by the sample code (QIAMEVXXX), while bacterial/viral coinfections are indicated in the footer of Table S2.
Additional corrections
- The registered trademark symbol “®” has been added where necessary. Specifically, we used "BioFire FilmArray®" (lines 54), instead of "BioFire FilmArray" and "QIAstat-Dx®" (lines 20, 69) instead of "QIAstat-Dx."
- The abbreviated form for parechovirus, HPeV, has been used where necessary.
- The layout of Figure 1 (page number 6) has been modified to comply with the MDPI figure template.
- In the Abstract (page number 1): lines 29-30; line 31; lines 36-37).
- In the Introduction (page number 2): line 69.
- In Materials and Methods: section 2.1. (page number 2; line 91); section 2.2. (page number 3; lines 127-139).
- In the Discussion (page number 7): lines 271-272.
- In the Conclusions (page number 9): lines 327-330.
- “LaLoQ” has been used instead of “LLoQ” of where necessary.
- “Log10” has been used instead of “log(10)” (lines 204 and 206).
- Supplementary Materials section (page number 9, lines 334-343).

Round 2
Reviewer 1 Report
Comments and Suggestions for Authors
The revised version of the manuscript includes the necessary corrections.